# Dynamic Risk Assessment of Voltage Violation in Distribution Networks with Distributed Generation

**DOI:** 10.3390/e25121662

**Published:** 2023-12-15

**Authors:** Wei Hu, Fan Yang, Yu Shen, Zhichun Yang, Hechong Chen, Yang Lei

**Affiliations:** Electric Power Science Research Institute, State Grid Hubei Electric Power Co., Ltd., Wuhan 430077, China; fan_yang0414@163.com (F.Y.); yu_shen0414@163.com (Y.S.); zhichun_yang0414@163.com (Z.Y.); hechong_chen@163.com (H.C.); leiyang1108@163.com (Y.L.)

**Keywords:** distributed generation, probabilistic load flow, risk assessment, boundary kernel density estimation, Rosenblatt inverse transform

## Abstract

In response to the growing demand for economic and social development, there has been a significant increase in the integration of distributed generation (DG) into distribution networks. This paper proposes a dynamic risk assessment method for voltage violations in distribution networks with DG. Firstly, considering the characteristics of random variables such as load and DG, a probability density function estimation method based on boundary kernel density estimation is proposed. This method accurately models the probability of random variables under different time and external environmental conditions, such as wind speed and global horizontal radiation. Secondly, to address the issue of correlated DG in the same region, an independent transformation method based on the Rosenblatt inverse transform is proposed, which enhances the accuracy of probabilistic load flow. Thirdly, a voltage violation severity index based on the utility function is proposed. This index, in combination with probabilistic load flow results, facilitates the quantitative assessment of voltage violation risks. Finally, the accuracy of the proposed method is verified on the IEEE-33 system.

## 1. Introduction 

To reduce the consumption of high-carbon-emission fossil fuels, the renewable energy sector has witnessed rapid growth. Photovoltaics (PVs) and wind turbines (WTs) have been extensively constructed, and they have been integrated into the distribution network in the form of distributed generation (DG), gradually transforming the conventional passive distribution network into a complex active distribution network [1,2,3].

Due to the significant influence of natural environmental factors such as solar radiation, PVs exhibit instability, marked by pronounced randomness and fluctuation [4]. Similarly, WTs are constrained by natural factors like wind speed, resulting in strong randomness and uncertainty [5]. The substantial integration of DG and the fluctuating loads introduce various sources of uncertainty into the distribution network system, significantly affecting its stable operation. This elevates the risk of voltage violations at network nodes, posing a considerable challenge to voltage control throughout the entire grid.

Hence, scholars have embarked on research pertaining to voltage violation risk assessment in distribution networks with DG [6,7,8]. However, the majority of these risk assessment studies primarily rely on historical data of DG to determine “static risks” of voltage violations at network nodes. They do not take into account load variations at different times and the influence of external environmental factors on DG. For instance, the load levels are typically lower during the late night compared to the afternoon, and the DG output can significantly differ under varying environmental conditions. These factors can lead to substantial variations in voltage violation risk in the same node at different times on the same day or at the same time on different days. Consequently, it becomes imperative to propose a dynamic risk assessment methodology for voltage violations in distribution networks with DG which can offer information for the coordination and control of network node voltages. Given the complex operating conditions in active distribution networks, deterministic load flow analysis methods often yield imprecise results, necessitating the use of probabilistic load flow (PLF) for analytical computations.

PLF analysis methods can be broadly categorized into three main types: the Monte Carlo Simulation (MCS) [9,10], the Point Estimate Method (PEM) [11,12], and analytical methods [13]. The MCS involves the simulation of numerous random variable samples through sampling, resulting in high accuracy but consuming a significant amount of time for calculations. The PEM approximates the moments of the output variables based on the characteristics of input random variables, but its computational efficiency is closely tied to the number of variables. Analytical methods rely on relationships among input random variables to calculate the probability statistical characteristics of output random variables. The key challenge of analytical methods lies in handling complex convolution calculations. Given the intricacies of convolution operations, it is possible to enhance computational efficiency by substituting convolution calculations with cumulant operations.

Currently, there are three main issues with using the probabilistic load flow based on the cumulant method (PLFCM) for dynamic risk assessment of voltage violations in distribution networks with DG.

The first issue pertains to the accuracy of the probability models for random variables such as load and DG, which directly affect the results of load flow calculations. There are currently two main approaches to describing probability models for random variables: parameter-based methods [14] and non-parameter-based methods [15]. Parameter-based methods begin by assuming that the research subject follows a certain probability distribution based on empirical observations. Then, they calculate the relevant parameter from actual samples to obtain a complete probability density function (pdf). However, due to the randomness of DG and the volatility of load, common probability density forms such as the Beta distribution and Weibull distribution often fail to accurately reflect the actual probability distribution, leading to significant errors. Non-parameter-based methods, on the other hand, do not require prior assumptions about the underlying probability distribution model or prior knowledge. Instead, they directly analyze the probability distribution based on actual data. Kernel density estimation (KDE) is the most widely used non-parameter method [16]. However, a notable characteristic of random variables like load and DG is that they are bounded. Traditional KDE methods, when applied to bounded data, can exhibit boundary effects that subsequently impact the results of probabilistic load flow. Additionally, another prominent feature of DG is its susceptibility to external environmental factors. As is well known, wind speed and global horizontal radiation are the most important factors affecting the output of WT and PV, respectively. How to achieve dynamic probity density estimation for DG remains an unresolved challenge under the influence of both.

The second issue pertains to the prerequisite that input variables must be mutually independent for the application of PLFCM. Due to the consistent characteristics of renewable energy sources in the same or neighboring areas, unit outputs often exhibit similar trends. The existence of correlations between input variables makes it impractical to directly employ this method for probabilistic load flow, necessitating the transformation of correlated outputs into independent ones. Presently, two widely used methods for achieving this are the Orthogonal inverse transform [17] and the Nataf inverse transform [18]. The Orthogonal inverse transform is straightforward and widely applied, while the Nataf inverse transform uses the correlation coefficient matrix, taking into account changes in equivalent correlation coefficients before and after transformation. However, both of these methods rely on the Pearson correlation coefficient and cannot capture nonlinear relationships among DGs. Moreover, they require variables to adhere to specific distribution types to achieve higher accuracy. Furthermore, the most challenging aspect of the Nataf inverse transform is determining the equivalent correlation coefficients. Although some scholars [19] have provided empirical formulas for a certain number of equivalent correlation coefficients for the reference of other researchers, they only cover a limited range of common probability distributions. Addressing how to achieve a more precise and widely applicable independent transformation for random variables is an issue that requires resolution.

The third issue is the establishment of a reasonable risk assessment framework, which is a key factor in achieving voltage violation risk assessment in distribution networks with DG. Risk is a comprehensive measure that combines the probability of an event occurring with the severity of its consequences [20]. When conducting risk assessments, both aspects should be taken into consideration. Traditional risk assessments often focus solely on the probability of a voltage violation occurring, without considering the severity of its consequences. Voltage violation events with a low probability but significant impact should receive more attention than those with a higher probability but less significant consequences. How to accurately portray the severity of losses resulting from risk events is a currently unresolved issue.

To address the aforementioned issues, this paper proposes a dynamic risk assessment method for voltage violations in distribution networks with DG. Firstly, a pdf estimation method is proposed, based on boundary kernel density estimation (BKDE), to overcome the problem of errors at boundary points when handling bounded data. Moreover, considering the significant impact of wind speed and global horizontal radiation on DG, conditional density is introduced to enable the dynamic probability density estimation of DG based on numerical weather prediction (NWP). Secondly, an independent transformation method based on the Rosenblatt inverse transform is proposed. This method accurately characterizes the correlations between DG variables, achieving an independent transformation of variables, and laying the foundation for subsequent PLFCM. Thirdly, a voltage violation severity index based on the utility function is proposed. By combining this index, an integrated risk assessment framework is constructed on the basis of the probability of voltage violations. This framework quantitatively analyzes the dynamic risk of voltage violations in distribution networks with DG at both the node and system levels. Finally, simulation tests on the IEEE-33 system validate the rationality and effectiveness of the proposed method.

## 2. Pdf Estimation Based on BKDE

KDE is a data-driven, non-parametric method for estimating pdf and has the advantage of being unaffected by the choice of prior models [21]. It is commonly used for data fitting when it is challenging to directly obtain the underlying pdf. This method is suitable for analyzing the probability characteristics of load and DG. Hence, in this paper, the KDE method is employed for estimation.

Given a set of independently and identically distributed data, *X*_1_, ..., *X_N_*, with an unknown density function *f*(*x*), the KDE is calculated as follows:(1)$${\hat{f}}_{X}\left(x\right)=\frac{1}{Nh}\sum_{i=1}^{N}K\left(\frac{x-X_{i}}{h}\right)$$
where *K*(∙) represents the kernel function; *N* represents the sample size; *h* represents the bandwidth; *x* represents the point at which the kernel density estimate is being calculated; and *X_i_* represents the *i*th sample.

The choice of the kernel function can indeed influence the results of KDE. Generally, the direction of the sample *X_i_* from the density point *x* does not affect the estimation, and the closer the distance from *x,* the more weight should be assigned to *X_i_*. Therefore, a unimodal kernel function centered at 0 is typically chosen. In this paper, the Epanechnikov kernel function is used:(2)$$K\left(z\right)=\frac{3}{4}\left(1-z^{2}\right)I\left(\left|z\right|\leq 1\right)$$
where *I*(∙) represents the indicator function, which has a value of 1 when *z* satisfies the condition and 0 otherwise. Given *z* = (*x* − *X_i_*)/*h* and substituting (2) into (1), we can obtain the complete formula for the KDE.

### 2.1. pdf Estimation for Load

Assuming in the distribution network there are *N_L_* load nodes, the load data for the distribution network at time *i*, denoted as $P_{i}^{L}$, can be expressed as follows:(3)$$P_{i}^{L}=\left[\begin{array}{cccc} P_{i1}^{L} & P_{i2}^{L} & \ldots & P_{iN_{L}}^{L} \end{array}\right]$$
where $P_{ik}^{L}$ represents the local load power measurement for node *k* at time *i*. Additionally, if there are historical data for the load power for a total of *T* time, the historical load data, *P_L_*, can be described as follows:(4)$$P^{L}=\left[\begin{array}{cccc} P_{11}^{L} & P_{12}^{L} & \ldots & P_{1N_{L}}^{L}\\ P_{21}^{L} & P_{22}^{L} & \ldots & P_{2N_{L}}^{L}\\ \vdots & \vdots & \vdots & \vdots \\ P_{T1}^{L} & P_{T2}^{L} & \ldots & P_{TN_{L}}^{L} \end{array}\right]$$

In (4), the data for each load column can be decomposed, based on daily patterns, into two components: the basic data with a daily cycle and the random fluctuation data. Therefore, when performing PLFCM, the pdf for node *k* at time *t* can be calculated using the power data set for different days at the same time *t*, denoted as $P_{t}^{L,k}$.
(5)$$P_{t}^{L,k}=P_{\left(mT_{d}+t\right)k}^{L},m=0,1,\ldots ,\frac{T}{T_{d}}-1$$
where *T_d_* represents the number of monitoring sample points within one day.

However, a significant characteristic of load is that it is bounded, and its power consumption is always greater than zero. Therefore, when performing KDE, it is necessary to consider the impact of boundary effects.

The expression for the bias of the KDE is as follows:(6)$$Bias\left[\hat{f}\left(x\right)\right]=\frac{h^{2}\sigma_{k}^{2}f^{\prime \prime }\left(x\right)}{2}+O\left(h^{4}\right)$$
where $\sigma_{k}^{2}$ = ∫*u*^2^*K*(*u*)*du*, and *K*(*u*) is the kernel function as shown in (2).

It can be observed that this bias diminishes as the bandwidth *h* decreases, approaching 0 at a rate of *h*^2^. However, when the pdf has a boundary at 0, the above expression is no longer applicable. Instead, it is replaced by:(7)$$E\left[{\hat{f}}_{K}\left(x\right)\right]=a_{0}\left(x\right)f\left(x\right)-ha_{1}\left(x\right)f^{\prime }\left(x\right)+O\left(h^{2}\right)$$
where *a_i_*(*x*) = $\int_{-1}^{x/h}$*u^i^K*(*u*)*du*. It can be observed that when *x* ≥ *h*, *a*_0_(*x*) = 1 and *a*_1_(*x*) = 0, leading to no difference in bias from (6). However, when 0 ≤ *x* < *h*, both *a*_0_(*x*) and *a*_1_(*x*) are non-zero, implying that bias is always present at the boundary.

In this case, when another kernel function *L*(*x*) is used to estimate *f*(*x*) once again, there is as follows:(8)$$E\left[{\hat{f}}_{L}\left(x\right)\right]=b_{0}\left(x\right)f\left(x\right)-hb_{1}\left(x\right)f^{\prime }\left(x\right)+O\left(h^{2}\right)$$
where *b*_0_ and *b*_1_ are similar to *a*_0_ and *a*_1_ from (7), except that (8) is specific to *L*(*x*). By performing a linear combination of (7) and (8), we can obtain:(9)$$b_{1}\left(x\right)*E\left[{\hat{f}}_{K}\left(x\right)\right]-a_{1}\left(x\right)*E\left[{\hat{f}}_{L}\left(x\right)\right]=\left[b_{1}\left(x\right)a_{0}\left(x\right)-a_{1}\left(x\right)b_{0}\left(x\right)\right]f\left(x\right)+O\left(h^{2}\right)$$

Equation (9) can be equivalently viewed as an estimation of *f*(*x*) using a new kernel function. This new kernel function is as follows:(10)$$K^{B}\left(x\right)=\frac{b_{1}\left(x\right)K\left(x\right)-a_{1}\left(x\right)L\left(x\right)}{b_{1}\left(x\right)a_{0}\left(x\right)-a_{1}\left(x\right)b_{0}\left(x\right)}$$

In particular, if *L*(*x*) is taken as *x***K*(*x*), *K^B^*(*x*) will have a simple form:(11)$$K^{B}\left(x\right)=\frac{\left(a_{2}\left(x\right)-a_{1}\left(x\right)x\right)K\left(x\right)}{a_{0}\left(x\right)a_{2}\left(x\right)-a_{1}^{2}\left(x\right)}$$

When *x* ≥ *h*, the new kernel function *K^B^*(*x*) is the same as *K*(*x*), but when 0 ≤ *x* < *h* (i.e., at the boundary), it adjusts the original kernel function. When *x* < 0, the estimated value is taken as 0.

Based on (1), (2), and (11), we can derive the calculation formula for BKDE by substituting (5), which provides the pdf for node *k* at time *t*.

Taking an exponential distribution with a parameter of 1 as an example, the performance of BKDE is tested. One thousand random numbers are generated from this exponential distribution, and pdf estimation is performed using both KDE and BKDE. Figure 1 shows the results of these two estimation methods compared to the actual pdf.

Observing Figure 1, it can be seen that, compared to the true density function, the results obtained by KDE exhibit a noticeable decrease at the boundary. In contrast, the results obtained by BKDE closely match the real situation, and the bias issues near the boundary have been significantly improved.

### 2.2. Conditional pdf Estimation for DG

For DG, its generated power is constrained between 0 and its maximum capacity. After normalization to the maximum capacity, its double boundary is [0, 1]. Therefore, when calculating the kernel function as shown in (11), the parameters *a_i_*(*x*) are:(12)$$a_{i}\left(x\right)=\int_{\left(x-1\right)/h}^{x/h}u^{i}K\left(u\right)du$$

In addition to its bounded nature, another significant characteristic of DG is its strong correlation with external factors. For instance, PVs might exhibit significant variations in power data at different times of one day due to differences in radiation. Therefore, the probability model for DG at time *t* can be characterized jointly by the conditional pdf and the predicted values of conditional variables at time *t*.

The conditional pdf for DG at time *t* can be represented as:(13)$${\hat{f}}_{P}\left(p_{t}\left|Y=y_{t}\right.\right)=\frac{{\hat{f}}_{P,Y}\left(p_{t},y_{t}\right)}{{\hat{f}}_{Y}\left(y_{t}\right)}$$
where *p_t_* is the DG output at time *t*, *y_t_* is the predicted value of the conditional variable at time *t*, ${\hat{f}}_{P}$ is the conditional pdf for DG, ${\hat{f}}_{Y}$ is the pdf for the conditional variable, and ${\hat{f}}_{P,Y}$ is the joint pdf.

Based on Equations (1), (2), and (11), the conditional pdf for DG can be represented as:(14)$${\hat{f}}_{P}\left(p_{t}\left|Y=y_{t}\right.\right)=\frac{\frac{1}{Nh_{1}h_{2}}\sum_{i=1}^{N}K_{1}^{B}\left(\frac{p_{t}-P_{i}}{h_{1}}\right)K_{2}^{B}\left(\frac{y_{t}-Y_{i}}{h_{2}}\right)}{\frac{1}{Nh_{2}}\sum_{i=1}^{N}K_{2}^{B}\left(\frac{y_{t}-Y_{i}}{h_{2}}\right)}$$
where *P_i_* represents historical samples of DG, *Y_i_* represents corresponding historical samples of conditional variables, *h*_1_ and *h*_2_ are bandwidth parameters, and $K_{1}^{B}$(∙) and $K_{2}^{B}$(∙) are kernel functions.

Different types of DG correspond to different conditional variables. In this paper, the conditional variable for PVs is global horizontal radiation, while for WTs, it is wind speed.

## 3. PLFCM Based on the Rosenblatt Inverter Transform

### 3.1. Independent Transform Based on Rosenblatt Inverter Transformation

#### 3.1.1. Joint pdf for DG

Due to the characteristics of wind and solar energy sources, DGs located in close geographical proximity exhibit some level of similarity in their output, showing spatial correlation. The higher the spatial correlation among DGs, the stronger the synchronization between different DGs, making PLFCM more challenging. Therefore, in order to establish a foundation for subsequent Rosenblatt inverse transform and PLFCM, it is essential to accurately characterize the joint pdf of DG.

Assuming that the multivariate data $P_{i1}^{D}$, ..., $P_{id}^{D}$ (*i* = 1, ..., *N*) are historical data from *d* different DGs ($P_{1}^{D}$, ..., $P_{d}^{D}$), their joint density function can be calculated through KDE using the following equation:(15)$${\hat{f}}_{P_{1}^{D}\ldots P_{d}^{D}}\left(p_{1}^{D},\ldots ,p_{d}^{D}\right)=\frac{1}{Nh_{1}\ldots h_{d}}\sum_{i=1}^{N}\prod_{j=1}^{d}K_{j}\left(\frac{p_{j}^{D}-P_{ij}^{D}}{h_{j}}\right)$$
where *h*_1_, ..., *h_d_* are bandwidth parameters, and *K_j_*(∙) are kernel functions corresponding to the variable $P_{j}^{D}$ (*j* = 1, ..., *d*).

Taking into account the boundary characteristics of DG and external conditional factors, the joint pdf for DG at time *t* can be represented as follows:(16)$${\hat{f}}_{P_{1}^{D},\ldots ,P_{d}^{D}}\left(p_{1t}^{D},\ldots ,p_{dt}^{D}\left|Y_{1}=y_{1t},\ldots ,Y_{d}=y_{dt}\right.\right)=\frac{\frac{1}{Nh_{1}^{p}\ldots h_{d}^{p}h_{1}^{y}\ldots h_{d}^{y}}\sum_{i=1}^{N}\prod_{j=1}^{d}K_{j}^{B}\left(\frac{p_{jt}^{D}-P_{ij}^{D}}{h_{j}^{p}}\right)\prod_{k=1}^{d}K_{k}^{B}\left(\frac{y_{kt}-Y_{ij}}{h_{k}^{y}}\right)}{\frac{1}{Nh_{1}^{y}\ldots h_{d}^{y}}\sum_{i=1}^{N}\prod_{k=1}^{d}K_{k}^{B}\left(\frac{y_{kt}-Y_{ij}}{h_{k}^{y}}\right)}$$

#### 3.1.2. Independent Transform

The Rosenblatt transform can directly convert a set of correlated non-normal variables *U^C^* = ($U_{1}^{C}$, $U_{2}^{C}$, ..., $U_{N}^{C}$)^T^ into independent standard normal variables *U^I^* = ($U_{1}^{I}$,$U_{2}^{I}$,...,$U_{I}^{n}$)^T^. According to the principle of equiprobability marginal transformation, it can be expressed as follows:(17)$$\left\{\begin{array}{l} \Phi \left(u_{1}^{I}\right)=F_{1}\left(u_{1}^{C}\right)\\ \Phi \left(u_{2}^{I}\right)=F_{2\left|1\right.}\left(u_{2}^{C}\left|u_{1}^{C}\right.\right)\\ \ldots \\ \Phi \left(u_{n}^{I}\right)=F_{n\left|1,2,\ldots ,n-1\right.}\left(u_{n}^{C}\left|u_{1}^{C},u_{2}^{C},\ldots ,u_{n-1}^{C}\right.\right) \end{array}\right.$$
where *Φ*(∙) represents a cumulative distribution function (CDF) of normal distribution, and *F*(∙) represents a conditional CDF.

Taking the inverse of (17) will yield the independent standard normal variables *U^I^*, which can be expressed as:(18)$$\left\{\begin{array}{l} u_{1}^{I}=\Phi^{-1}\left[F_{1}\left(u_{1}^{C}\right)\right]\\ u_{2}^{I}=\Phi^{-1}\left[F_{2\left|1\right.}\left(u_{2}^{C}\left|u_{1}^{C}\right.\right)\right]\\ \ldots \\ u_{n}^{I}=\Phi^{-1}\left[F_{n\left|1,2,\ldots ,n-1\right.}\left(u_{n}^{C}\left|u_{1}^{C},u_{2}^{C},\ldots ,u_{n-1}^{C}\right.\right)\right] \end{array}\right.$$

Equation (18) is known as the Rosenblatt transform. It is not influenced by the distribution type or the correlation type and is considered a precise transformation method. The conditional CDF can be obtained by integrating the joint PDF derived from (16).

Through its inverse transform, standard normal variables can be transformed into independent samples of DG, and the basic steps are as follows:

(1) Generate *m* samples that follow the standard normal distribution functions $U_{1}^{I}$, $U_{2}^{I}$, ..., $U_{n}^{I}$.

(2) Use (18) to obtain:(19)$$u_{1}^{C}=F_{1}^{-1}\left[\Phi \left(u_{1}^{I}\right)\right]$$

From (19), it can obtain *m* samples of one of the DG, denoted as $u_{1}^{C}$.

(3) Extend to *n* DG, the following is applicable:(20)$$u_{n}^{C}=F_{n\left|1,2,\ldots ,n-1\right.}^{-1}\left[\Phi \left(u_{n}^{I}\right)\left|u_{1}^{C},u_{2}^{C},\ldots ,u_{n-1}^{C}\right.\right]$$

### 3.2. PLFCM

#### 3.2.1. Linearized Load Flow Model

Considering the random variation in node injection power, the polar form of the system power flow equation is Taylor-expanded at the base operating point, retaining only the first-order term, yielding as follows:(21)$$\left\{\begin{array}{l} \Delta X=J_{0}^{-1}\Delta W=S_{0}\Delta W\\ \Delta Z=G_{0}J_{0}^{-1}\Delta W=T_{0}\Delta W \end{array}\right.$$
where ∆*X*, ∆*W*, and ∆*Z* represent the perturbations in node state variables, node injection power variables, and branch power variables, respectively. *S*_0_ and *T*_0_ are sensitivity matrices. *J*_0_ is the Jacobian matrix obtained in the last iteration of the power flow calculation. *G*_0_ = (*∂Z*/*∂X*)|*_X_*_=*X*0_.

#### 3.2.2. Cumulant Computation

Cumulants can be calculated based on origin moments that do not exceed their order, and in this paper, only the first eight orders are considered. The relationship between various orders of cumulants and origin moments, as well as the specific derivation process, can be found in [22].

For load, the origin moments of all orders can be calculated using the density function obtained from its historical data based on BKDE.
(22)$$\alpha_{v}=\int_{-\infty }^{+\infty }x^{v}\hat{f}\left(x\right)dx$$
where *α_v_* represents the *v*th order origin moment.

For DG variables with correlation, it is necessary to first generate mutually independent samples following the standard normal distribution. Then, using the Rosenblatt inverse transform, independent samples of the DG variables can be obtained. Based on this, the cumulants of that variable can be calculated using the relationship between origin moments and cumulants.

Once the cumulants of the node injection power changes, Δ*W*, are computed, it can then determine the cumulants of the node state changes, Δ*X*, and the cumulants of the branch power changes, ΔZ, for each order using the following equation:(23)$$\left\{\begin{array}{l} \Delta X^{\left(k\right)}=S_{0}^{\left(k\right)}\left(\Delta W_{DG}^{\left(k\right)}+\Delta W_{L}^{\left(k\right)}\right)\\ \Delta Z^{\left(k\right)}={\left(G_{0}J_{0}^{-1}\right)}^{\left(k\right)}\left(\Delta W_{DG}^{\left(k\right)}+\Delta W_{L}^{\left(k\right)}\right) \end{array}\right.$$
where Δ$W_{DG}^{\left(k\right)}$ and Δ$W_{L}^{\left(k\right)}$ represent the *k*th order cumulants for changes in DG injection power and changes in load injection power, respectively.

#### 3.2.3. Cornish–Fisher Series Expansion

Series expansion can approximate the cumulants of output variables as probability distributions. According to [23], the Cornish–Fisher series provides higher accuracy. Therefore, in this study, the CDF of output variables is obtained using the Cornish–Fisher series.

The approach of the Cornish–Fisher series expansion entails initially selecting a specific quantile, followed by computing the quantiles of the state variable, and ultimately deriving the cumulative distribution of that variable. Assuming *α* represents the quantile of random variable *X*, *x*(*α*) can be expressed as follows:(24)$$x(\alpha )=\zeta (\alpha )+\frac{\zeta^{2}(\alpha )-1}{6}g_{3}+\frac{\zeta^{3}(\alpha )-3\zeta (\alpha )}{24}g_{4}-\frac{2\zeta^{3}(\alpha )-5\zeta (\alpha )}{36}g_{3}^{2}+\frac{\zeta^{4}(\alpha )-6\zeta^{2}(\alpha )+3}{120}g_{5}+\ldots$$
where *ζ*(*α*) = *Φ*^−1^(*α*), and *g_v_* represents the *v*th order normalized cumulant. 

By utilizing the relationship *x*(*α*) = *F*^−1^(*α*), the CDF *F*(*x*) of the output random variable *X* can be determined, thereby providing the probability of node voltage violation.

## 4. The Voltage Violation Risk Assessment Metric Based on the Utility Function

This section provides a comprehensive assessment of node voltage violation risk in distribution networks with DG by considering both the probability of voltage violation and the severity of such violation. This approach allows for a quantitative evaluation at both the node and system levels, serving as the basis for coordinated voltage control.

### 4.1. The Probability of Voltage Violation

The voltage violation probability refers to the likelihood of nodes deviating from the permissible voltage range, encompassing both overvoltage and undervoltage conditions. It can be determined through the previously conducted PLFCM analysis.
(25)$$r\left(V_{i}\right)=\left\{\begin{array}{l} F\left(V_{i}^{min}\right),\;\;\;\;V_{i}<V_{i}^{min}\\ 0,\;\;\;\;\;\;\;\;\;V_{i}^{min}<V_{i}<V_{i}^{max}\\ 1-F\left(V_{i}^{max}\right),\;\;\;\;V_{i}>V_{i}^{max} \end{array}\right.$$
where *r*(*V_i_*) represents the voltage violation probability for node *i*, *F*(*V_i_*) represents the voltage CDF for node *I*, and $V_{i}^{min}$*,* and $V_{i}^{max}$ represent the lower and upper voltage limits permissible for node *i*, respectively.

### 4.2. The Severity of Voltage Violation

Voltage violation severity reflects the level of severity caused to the system and equipment when the voltage deviates from permissible values. The traditional voltage violation severity function, denoted as *S_e_*, is represented by linear functions.
(26)$$S_{e}=\left\{\begin{array}{l} \frac{\left(V_{i}-V_{i}^{max}\right)}{V_{i}^{max}},V_{i}>V_{i}^{max}\\ \frac{\left(V_{i}^{min}-V_{i}\right)}{V_{i}^{min}},V_{i}<V_{i}^{min} \end{array}\right.$$

In reality, many electrical devices exhibit more severe consequences as voltage deviation increases. For instance, a slight voltage deviation from the permissible range might lead to a decrease in product quality in manufacturing equipment, and as the extent of voltage deviation deepens, it can even impact equipment safety, resulting in significant consequences such as equipment damage. Therefore, the severity assessment function should be more sensitive to reflect the consequences when the deviation is severe. This paper employs the utility function to define the severity function of node voltage violation.
(27)$$S_{e}\left(\theta \left(V_{i}\right)\right)=\left(\frac{e^{k\theta \left(V_{i}\right)}-1}{e-1}\right)$$
where *k* represents the risk factor, and a larger value indicates greater sensitivity to risk. *θ*(*Vi*) represents the voltage deviation index, defined as follows:(28)$$\theta \left(V_{i}\right)=\left\{\begin{array}{l} \frac{V_{i}^{min}-V_{i}}{V_{B}},\;\;V_{i}<V_{i}^{min}\\ 0,\;\;\;\;V_{i}^{min}<V_{i}<V_{i}^{max}\\ \frac{V_{i}-V_{i}^{max}}{V_{B}},V_{i}>V_{i}^{max} \end{array}\right.$$

### 4.3. The Comprehensive Assessment of Voltage Violation

Taking into account both voltage violation probability and the severity of such violations, the comprehensive risk index for voltage violation of node *i*, *R_i_*, and the system-level comprehensive risk index *R_s_*, can be defined as follows:(29)$$R_{i}=\int_{-\infty }^{V_{i}^{min}}f\left(V_{i}\right)\cdot S_{e}\left(\theta \left(V_{i}\right)\right)dV_{i}+\int_{V_{i}^{max}}^{+\infty }f\left(V_{i}\right)\cdot S_{e}\left(\theta \left(V_{i}\right)\right)dV_{i}$$
(30)$$R_{s}=\sum_{i=1}^{n}R_{i}$$
where *n* represents the number of system nodes, and *f*(*V_i_*) represents the voltage pdf of node *i*, which can be derived from *F(V_i_*) using numerical differentiation.

## 5. The Process of the Proposed Method

The method proposed in this paper enables dynamic risk assessment of voltage violation in distribution networks with DG, which is crucial for coordinated voltage control. The specific flow chart of the method is depicted in Figure 2 and comprises three main steps:

Step 1: Probability Density Function Estimation. Accurate pdfs are obtained through BKDE, utilizing historical data for both load and DG, while also accounting for the influence of external factors.

Step 2: Probability Load Flow Calculation. Considering the correlation between DGs, independent DG variables are obtained through the Rosenblatt inverse transform. Based on these variables, PLFCM is applied to compute the probability distribution of node voltages.

Step 3: Risk Assessment. The severity of voltage violation at the node is considered through the application of the utility function, and this, combined with the results of PLFCM, yields dynamic risk assessment outcomes at both the node and system levels.

## 6. Case Study

Due to the confidentiality requirement, real grid data are difficult to obtain, so the case study in this paper is conducted using an improved IEEE-33 system, as depicted in Figure 3. A WT with a capacity of 600 kW is connected to node 26, while two PVs are connected to node 33 and node 15 with capacities of 500 kW and 450 kW, respectively. The daytime DG penetration rate is approximately 30%. The output data for WTs and PVs is real and derived from a WT and PVs located in a specific region in northwest China, along with historical weather records providing the corresponding wind speed and global horizontal radiation data. To better simulate the probability density of the 24 h base load, the ratio of the load for each hour relative to the maximum load of that node is defined according to the solid line in Figure 4. In accordance with daily patterns, the load can be decomposed into two components: periodic base data and random fluctuation data. To reflect the load’s volatility, the standard deviation of the load is set to 20%. Following the three-sigma rule, it can be considered that the load ratios at each time fall within the shaded area of Figure 4.

In this section, the accuracy of pdf modeling under real data using BKDE and KDE is compared. Subsequently, the effectiveness of the proposed method for risk assessment is validated through simulations in two cases: when there is abundant radiation and high wind speed resulting in higher DG output during the daytime, and when there is no radiation at night coupled with low wind speed leading to reduced DG output.

### 6.1. The Performance of pdf Modeling

Section 2.1 compares the pdf modeling results of BKDE and KDE with simulation data through a simple example. Here, the accuracy of the modeling of the two is compared again through real DG data. Since the real pdf of the DG data is unknown, the relative error between the mean and variance of the modeled pdf and the mean and variance of the real measured data is used as the criterion for comparison. When using the measured data for PV, the data for the time when the PV is not producing power at night were removed. The results are presented in Table 1. Figure 5, Figure 6 and Figure 7 show histograms of the DG real data as well as the pdfs calculated by BKDE and KDE. 

The results from Figure 5, Figure 6 and Figure 7 reveal that KDE exhibits a segment of density curve near the boundary points that markedly contradicts the actual histogram trend, whereas BKDE yields results consistent with the actual trend. It can be affirmed that the results from BKDE are more in line with the actual data-based histogram compared to KDE.
(31)$$RE=\frac{\left|Value_{c}-Value_{r}\right|}{Value_{r}}\times 100\%$$
where *Value_c_* represents the mean or variance of the modeled pdf, and *Value_r_* represents the mean or variance of the real measured data.

### 6.2. Case 1: High DG Output

Assuming that, based on NWP, the wind speed forecast for WT at 11 a.m. on a certain day is 10 m/s, and the global horizontal radiation at PV1 is 850 W/m^2^, while at PV2, it is 900 W/m^2^. Based on the BKDE, the pdfs and CDFs of WT, PV1, and PV2 can be obtained, as shown in Figure 8, along with the joint pdf and joint CDF between these variables, as illustrated in Figure 9, Figure 10 and Figure 11.

From the above figures, it is evident that the variables WT, PV1, and PV2 do not satisfy the condition of mutual independence between them. Therefore, it is not possible to directly perform PLFCM, and an independent transformation is required.

After performing the Rosenblatt inverse transformation, the correlated DG variables were transformed to become independent samples. Table 2 and Table 3 present the Pearson correlation coefficient, Kendall correlation coefficient, and Spearman correlation coefficient between WT, PV1, and PV2 before and after the transformation. The results indicate that after the transformation, the absolute values of the correlation coefficients between DG variables were largely reduced to below 0.1, significantly decreasing their correlations and rendering them nearly independent.

In this paper, the accuracy of the proposed method is validated through the MCS method for PLF. Additionally, a method based on Orthogonal inverse transform is employed to calculate PLF for comparative purposes. To account for the correlation, the MCS method used DG data that originally had correlations before the independence transformation. It employed random sampling based on the joint pdf of the DG variables obtained previously. 

In total, the MCS method conducted 20,000 random samples, performed deterministic load flow, and obtained the pdf of node voltages. Figure 12 illustrates the voltage pdf of three DG-connected nodes and the last node 18.

Due to space limitations, it is difficult to plot the pdf of all nodes under the three different methods. Therefore, in this paper, the results of the MCS method are used as the standard to calculate the relative errors of the mean and variance obtained by both the PLFCM with Rosenblatt inverse transform and PLFCM with Orthogonal inverse transform. The results are shown in Table 4.

Observing Figure 12, it can be noted that the results obtained from the PLFCM based on BKDE and Rosenblatt inverse transformation are in good agreement with the outcomes of the MCS method. Node 26 is located in the middle of the radial distribution network and has a WT connection, which is why its voltage remains within the allowable range without violations. However, nodes 33, 15, and 18 are situated at the endpoints of the distribution network. Despite having DG support at these nodes or in their vicinity, voltage violations below the lower limit still occur due to the impedance losses in the transmission lines.

Moreover, as can be seen from Table 4, the error of PLFCM based on the Orthogonal inverse transform is significantly larger than that of PLFCM based on the Rosenblatt inverse transform proposed in this paper. This is due to the fact that the Orthogonal inverse transform is unable to characterize the nonlinear relationship that exists between the DG variables.

The time used for the calculation of the three methods is shown in Table 5.

By comparing the computation times in Table 5, it can be found that the computation time of PLFCM with Rosenblatt inverse transform and PLFCM with Orthogonal inverse transform is greatly reduced and the computation efficiency is improved compared with the MCS method. Through the above analysis, in general, the method proposed in this paper has high accuracy and prediction precision, and the calculation time is greatly reduced.

The corresponding risk indexes can be calculated based on the node voltage violation probability combined with the severity of the violation. A risk factor of three is assigned to regular nodes, while a risk factor of four is assigned to DG-connected nodes. Table 6 presents the voltage violation probability and risk indexes for each node. Nodes with risk indexes above 0.01 are considered high risk, while those between 0.0001 and 0.01 are categorized as medium risk, values below 0.0001 are classified as low risk, and a risk index of 0 indicates no risk. Figure 13 illustrates the risk zones of the system at different levels at 11 a.m.

### 6.3. Case 2: Low DG Output

Assuming that, based on NWP, the wind speed forecast for WT at 9 p.m. on a certain day is 6 m/s. Since it is night-time, the global horizontal radiation is zero, resulting in no power output from PV1 and PV2. Utilizing the BKDE, the pdf and CDF for WT can be obtained, as shown in Figure 14. 

As there is only one DG generating power at this moment, there is no correlation among multiple DGs. Hence, the PLFCM can be performed directly as there is no need for independent transformation. So in this section, only two methods are used for calculating the pdf of voltage, which are PLFCM and MCS. The sampling number of the MCS is still 20,000 times. The pdf of voltage at the WT-connected node and the last node 18 are shown in Figure 15. Table 7 illustrates the relative errors of the mean and variance of each node. Table 8 demonstrates the computation time for both methods.

From Figure 15, it can be observed that despite the relatively low WT output, node 26, due to its location and reduced load, maintains its voltage within the normal range. However, the end-node 18 experiences consistently low voltage levels during the night due to the lack of nearby PV power support. The voltage violation probability and risk indexes for each node at this time are presented in Table 9. Figure 16 illustrates the risk levels of the system at 9 p.m.

## 7. Conclusions

A dynamic risk assessment method for voltage violation in distribution networks with DG is proposed in this paper, which mainly contributes to the following three points.

Firstly, a pdf modeling method based on BKDE and conditional density is proposed. This method provides a more accurate representation of stochastic variables with boundary constraints, such as load and DG. Simulation results on actual DG data indicate that the relative errors of mean and variance using this method are approximately one-third of those obtained with KDE.

Secondly, to address the correlation between DG variables within the same region, an independent transformation method based on the Rosenblatt inverter transformation is proposed. This method has the capability to reduce the absolute values of correlation coefficients between random variables to below 0.1, laying the foundation for accurate PLFCM. Compared to the PLFCM based on the Orthogonal inverse transform, the proposed method demonstrates a reduction of one to two orders of magnitude in the relative errors of mean and variance, with comparable computational times.

Finally, an index for the severity of voltage violation, based on a utility function, is proposed. This index, combined with the voltage violation probability, quantitatively characterizes the risk of voltage violation at nodes and provides essential information for coordinated voltage control.

## Figures and Tables

**Figure 1 entropy-25-01662-f001:**
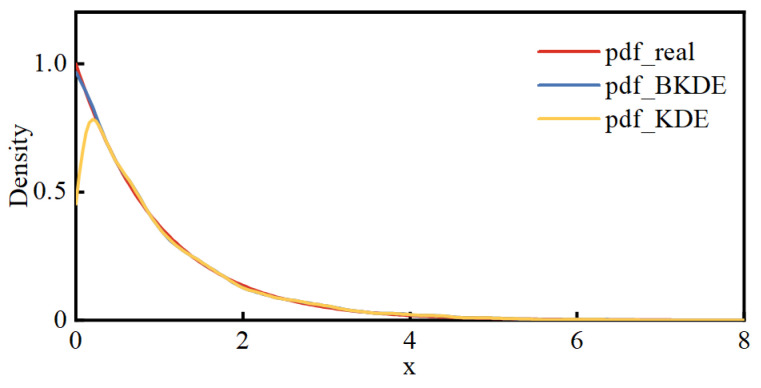
Results of the two estimation methods with the actual pdf.

**Figure 2 entropy-25-01662-f002:**
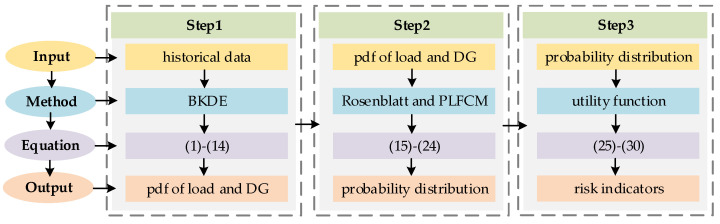
The flow chart of the proposed method.

**Figure 3 entropy-25-01662-f003:**
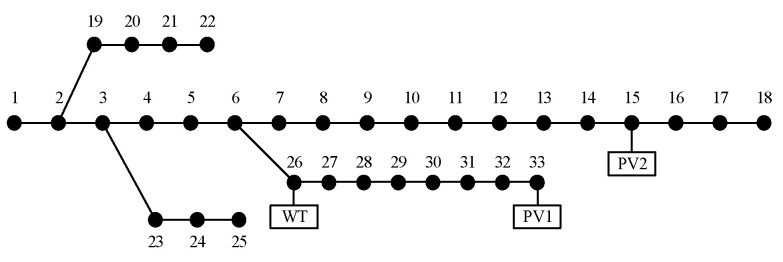
Improved IEEE-33 system.

**Figure 4 entropy-25-01662-f004:**
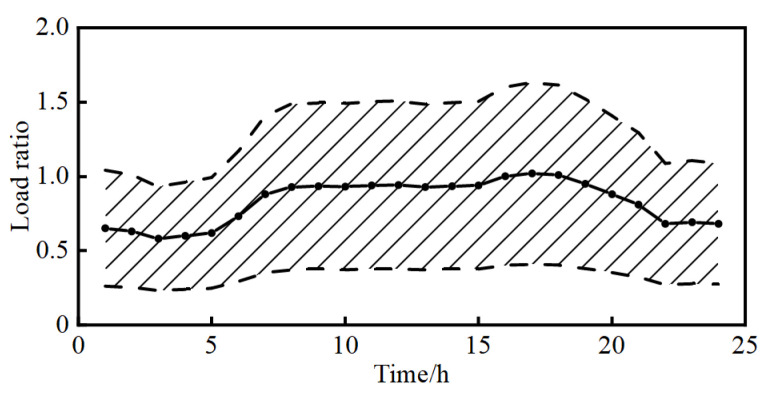
Load ratio at different times.

**Figure 5 entropy-25-01662-f005:**
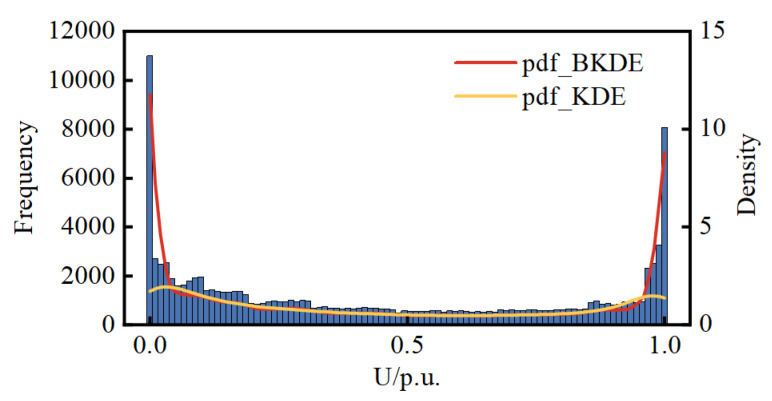
Histograms of the WT and the pdfs calculated by BKDE and KDE.

**Figure 6 entropy-25-01662-f006:**
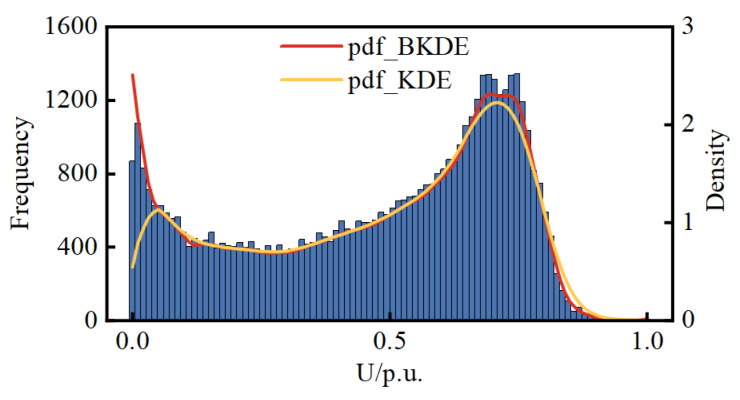
Histograms of the PV1 and the pdfs calculated by BKDE and KDE.

**Figure 7 entropy-25-01662-f007:**
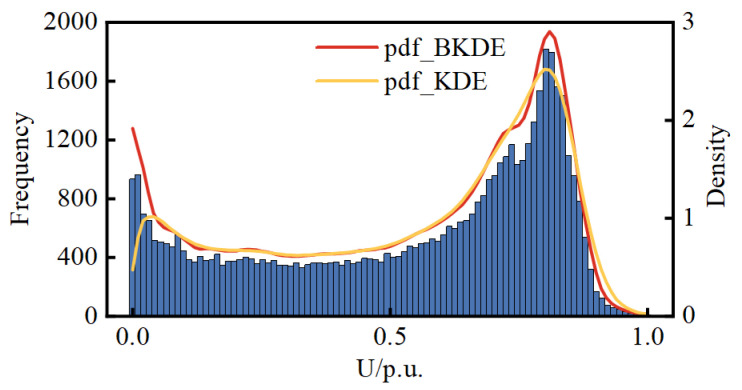
Histograms of the PV2 and the pdfs calculated by BKDE and KDE.

**Figure 8 entropy-25-01662-f008:**
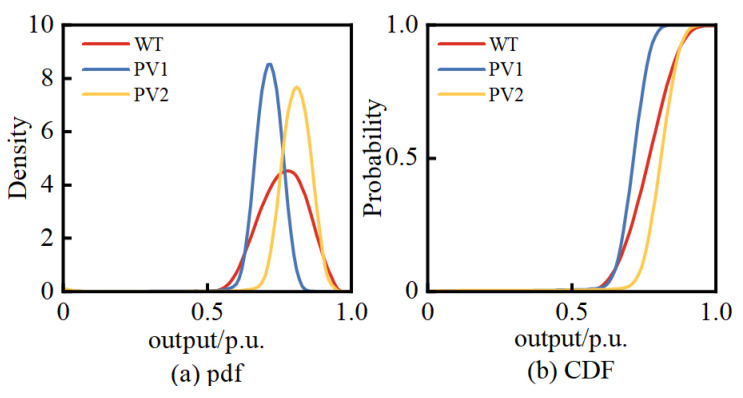
The pdfs and CDFs of WT, PV1, and PV2.

**Figure 9 entropy-25-01662-f009:**
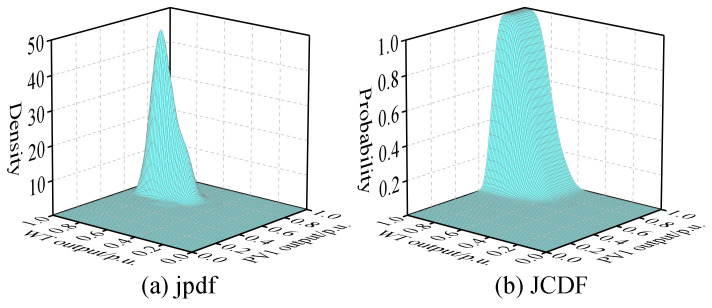
The joint pdf and joint CDF between WT and PV1.

**Figure 10 entropy-25-01662-f010:**
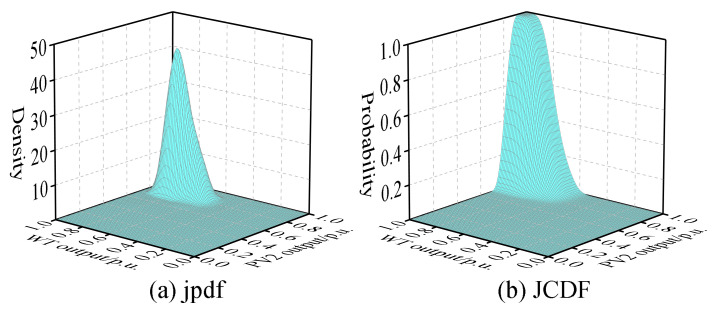
The joint pdf and joint CDF between WT and PV2.

**Figure 11 entropy-25-01662-f011:**
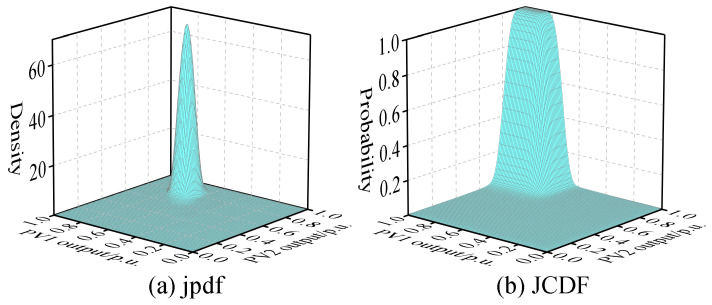
The joint pdf and joint CDF between PV1 and PV2.

**Figure 12 entropy-25-01662-f012:**
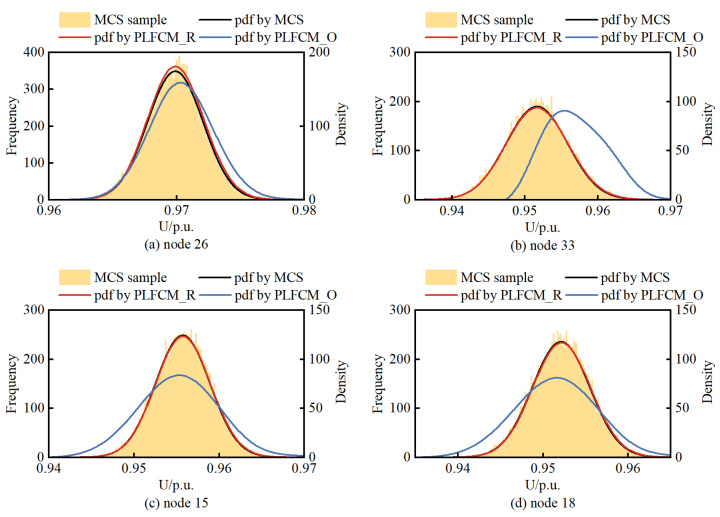
The voltage pdf of node. (**a**) node 26. (**b**) node 33. (**c**) node 15. (**d**) node 18.

**Figure 13 entropy-25-01662-f013:**
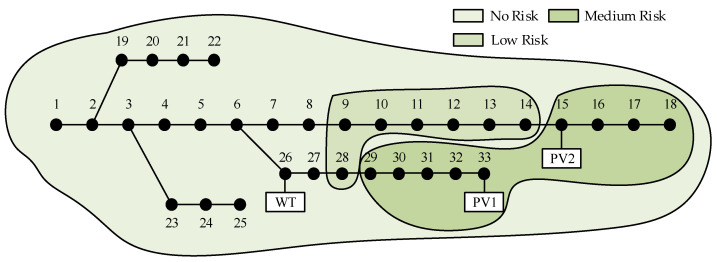
Risk level chart for system in case 1.

**Figure 14 entropy-25-01662-f014:**
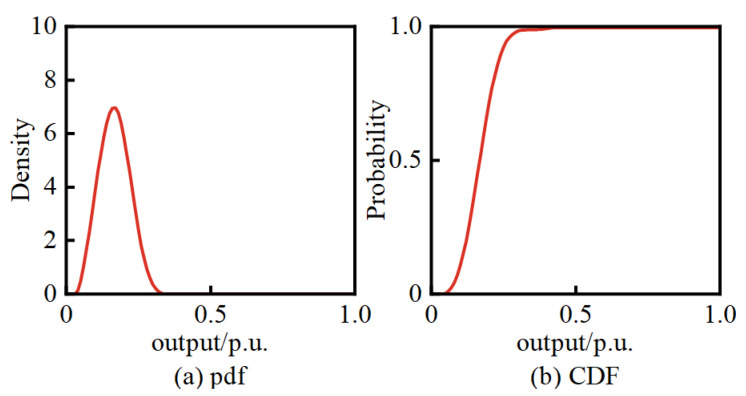
The pdf and CDF of WT.

**Figure 15 entropy-25-01662-f015:**
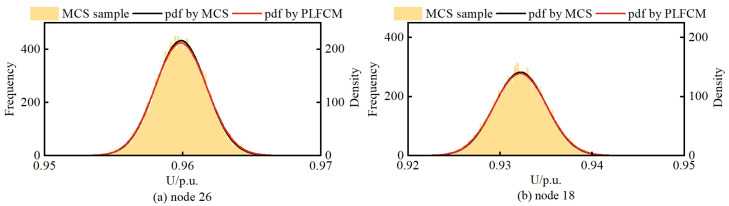
The voltage pdf of node. (**a**) node 26. (**b**) node 18.

**Figure 16 entropy-25-01662-f016:**
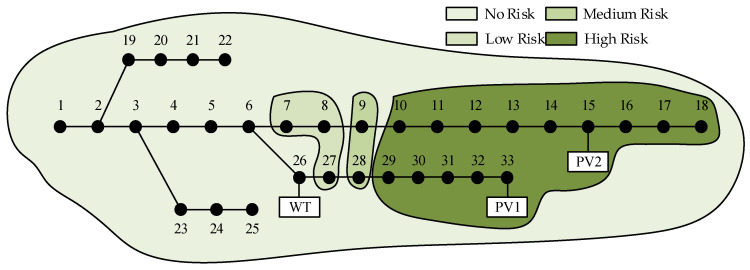
Risk level chart for system in case 2.

**Table 1 entropy-25-01662-t001:** The relative errors of BKDE and KDE.

	BKDE		KDE	
	RE of Mean (%)	RE of Variance (%)	RE of Mean (%)	RE of Variance (%)
**WT**	5.35	10.20	15.55	27.43
**PV1**	0.070	2.42	0.23	7.96
**PV2**	0.051	2.98	0.18	8.07

**Table 2 entropy-25-01662-t002:** Correlation coefficients before transformation.

	Pearson	Kendall	Spearman
**WT/PV1**	−0.2913	−0.2144	−0.2922
**WT/PV2**	−0.2882	−0.2126	−0.2896
**PV1/PV2**	0.9926	0.9511	0.9937

**Table 3 entropy-25-01662-t003:** Correlation coefficients after transformation.

	Pearson	Kendall	Spearman
**WT/PV1**	−0.0873	−0.1193	−0.0800
**WT/PV2**	−0.0925	−0.0625	−0.0929
**PV1/PV2**	0.1256	0.0795	0.1004

**Table 4 entropy-25-01662-t004:** The relative errors of PLFCM with Rosenblatt inverse transform and PLFCM with Orthogonal inverse transform.

Node No.	PLFCM with Rosenblatt Inverse Transform		PLFCM with Orthogonal Inverse Transform	
	RE of Mean (%)	RE of Variance (%)	RE of Mean (%)	RE of Variance (%)
2	6.1626 × 10^−5^	0.8711	0.0012	22.0573
3	3.3372 × 10^−4^	1.1296	0.0078	23.3937
4	0.0016	0.5352	0.0159	30.5238
5	0.0027	0.3139	0.0247	34.2327
6	0.0047	0.2804	0.0404	32.7683
7	0.0043	0.2230	0.0329	34.4186
8	0.0042	0.4460	0.0119	33.8826
9	0.0039	0.5258	0.0118	30.8185
10	0.0039	0.5358	0.0288	37.4144
11	0.0038	0.5457	0.0311	30.5229
12	0.0038	0.5538	0.0350	36.3805
13	0.0040	0.5333	0.0437	36.2802
14	0.0045	0.4495	0.0448	32.3938
15	0.0052	0.4262	0.0406	32.8715
16	0.0058	0.3688	0.0447	37.9928
17	0.006	0.3173	0.0509	33.5184
18	0.0072	0.3027	0.0531	31.7283
19	1.2526 × 10^−4^	1.1032	0.0012	18.1156
20	8.4721 × 10^−4^	2.6008	0.0018	6.5230
21	8.8285 × 10^−4^	2.4950	0.0019	5.9387
22	8.6273 × 10^−4^	2.8694	0.0018	4.9922
23	6.0421 × 10^−4^	2.2021	0.0059	14.0596
24	0.0024	2.9833	0.0040	7.3964
25	0.0039	2.8185	0.0025	5.7725
26	0.0053	0.1665	0.0525	31.4057
27	0.0058	0.3903	0.0744	30.1177
28	0.0065	1.2671	0.1548	22.8691
29	0.0065	1.6573	0.2179	19.4942
30	0.0070	1.6806	0.2693	18.1796
31	0.0080	1.1913	0.4310	7.4600
32	0.0082	1.0459	0.4991	1.9288
33	0.0083	0.9570	0.5981	22.9691

**Table 5 entropy-25-01662-t005:** Computation time of three methods.

Method	MCS of 20,000 Times	PLFCM with Rosenblatt Inverse Transform	PLFCM with Orthogonal Inverse Transform
**Time (s)**	305.46	2.15	2.06

**Table 6 entropy-25-01662-t006:** The voltage violation probability and risk indexes for the nodes in case 1.

Node No.	Probability (%)	Indexes	Node No.	Probability (%)	Indexes
2	0	0	19	0	0
3	0	0	20	0	0
4	0	0	21	0	0
5	0	0	22	0	0
6	0	0	23	0	0
7	0	0	24	0	0
8	0	0	25	0	0
9	0.001	3.412 × 10^−9^	26	0	0
10	0.0512	3.342 × 10^−7^	27	0	0
11	0.087	6.445 × 10^−7^	28	0.072	4.723 × 10^−7^
12	0.185	1.705 × 10^−6^	29	10.888	3.293 × 10^−4^
13	2.199	3.892 × 10^−5^	30	25.763	1.094 × 10^−3^
14	4.307	9.117 × 10^−5^	31	39.445	2.087 × 10^−3^
15	4.263	1.236 × 10^−4^	32	40.484	2.195 × 10^−3^
16	9.208	2.405 × 10^−4^	33	35.688	2.451 × 10^−3^
17	22.188	7.548 × 10^−4^	system	\	0.0104
18	27.372	1.010 × 10^−3^			

**Table 7 entropy-25-01662-t007:** The relative errors of PLFCM.

Node No.	RE of Mean (%)	RE of Variance (%)	Node No.	RE of Mean (%)	RE of Variance (%)
2	4.7903 × 10^−4^	1.5650	18	1.2851 × 10^−4^	2.1548
3	0.0030	1.7234	19	5.5308 × 10^−4^	1.5532
4	0.0029	2.8588	20	1.3169 × 10^−4^	2.8045
5	0.0029	1.9286	21	3.2046 × 10^−4^	2.9817
6	0.0014	2.3425	22	3.7397 × 10^−4^	3.1784
7	0.0014	3.1850	23	0.0055	1.7064
8	0.0013	2.6184	24	0.0109	1.8469
9	8.6506 × 10^−4^	2.5399	25	0.0148	2.3036
10	5.2550 × 10^−4^	1.4705	26	0.0012	2.4474
11	5.0435 × 10^−4^	2.4533	27	0.0011	2.5679
12	4.6079 × 10^−4^	2.3993	28	0.0014	2.1355
13	7.9488 × 10^−4^	3.2392	29	0.0042	2.3226
14	0.0013	3.2261	30	0.0049	1.3542
15	9.9030 × 10^−4^	2.2084	31	0.0033	2.1923
16	6.5868 × 10^−4^	3.1785	32	0.0031	2.1712
17	3.5136 × 10^−4^	2.1632	33	0.0031	2.1663

**Table 8 entropy-25-01662-t008:** Computation time of two methods.

Method	MCS of 20,000 Times	PLFCM
**Time (s)**	133.66	2.07

**Table 9 entropy-25-01662-t009:** The voltage violation probability and risk indexes for the nodes in case 2.

Node No.	Probability (%)	Indexes	Node No.	Probability (%)	Indexes
2	0	0	19	0	0
3	0	0	20	0	0
4	0	0	21	0	0
5	0	0	22	0	0
6	0	0	23	0	0
7	0.0004	1.945 × 10^−9^	24	0	0
8	0.1	1.075 × 10^−5^	25	0	0
9	56.672	1.819 × 10^−3^	26	0	0
10	98.612	0.0101	27	0.009	4.600 × 10^−8^
11	99.3316	0.0103	28	68.155	3.263 × 10^−3^
12	99.831	0.0121	29	99.300	0.0139
13	100	0.0209	30	99.916	0.0190
14	100	0.0241	31	99.997	0.0250
15	100	0.0352	32	100	0.0263
16	100	0.0281	33	100	0.0359
17	100	0.0311	system	\	0.329
18	100	0.0320			

## Data Availability

The data presented in this study are available on request from the corresponding author.

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
