# Peer review of "Dynamic Risk Assessment of Voltage Violation in Distribution Networks with Distributed Generation"

_entropy, 2023, doi:10.3390/e25121662_

Round 1

Reviewer 1 Report

Comments and Suggestions for Authors

The paper addresses some challenges in the context of voltage violation risk assessment in distribution networks with distributed generation (DG). The issues pertaining to probability modeling, achieving independence among input variables, and the establishment of a comprehensive risk assessment framework are well-articulated. As computational complexity is a concern for real-time Voltage Violation Risk Assessment, it would be helpful to include a discussion on the computational overhead of the proposed approach. Besides, the numerical example using real data is interesting. To benchmark the proposed approach, it will be useful to compare with existing algorithm(s).

Comments on the Quality of English Language

The English usage of this manuscript needs improvement. There are typos and grammar mistakes.

Reviewer 2 Report

Comments and Suggestions for Authors

This paper proposes a method to evaluate the risk of voltage violations in distributed networks with DG. The technique considers different time and external environmental conditions based on weather forecasts. First, a probability density function (PDF) method based on boundary kernel density estimation (BKDE) is proposed using historical data. Then, independent DG variables are obtained through the Rosenblatt inverse transform. Finally, a voltage violation severity index based on the utility function is proposed. As a case study, simulations are performed in the IEEE-33 system in two scenarios: high DG generation and low DG generation.

The motivation of this field of study is well explained to the readers. However, my main concern is that the simulations presented just exemplify the proposed methodology and do not clearly show how other methodologies previously introduced in the state of the art are improved. To solidly argue the article's contribution, choosing a benchmark against which a clear improvement can be demonstrated is essential. 

Other important aspects to improve are the following:

·         If possible, improve the quality of Figures 6, 7, and 8.

·         The conclusions must be rewritten, clearly identifying the contribution. It is also suggested to include some quantitative results.

Comments on the Quality of English Language

This paper proposes a method to evaluate the risk of voltage violations in distributed networks with DG. The technique considers different time and external environmental conditions based on weather forecasts. First, a probability density function (PDF) method based on boundary kernel density estimation (BKDE) is proposed using historical data. Then, independent DG variables are obtained through the Rosenblatt inverse transform. Finally, a voltage violation severity index based on the utility function is proposed. As a case study, simulations are performed in the IEEE-33 system in two scenarios: high DG generation and low DG generation.

The motivation of this field of study is well explained to the readers. However, my main concern is that the simulations presented just exemplify the proposed methodology and do not clearly show how other methodologies previously introduced in the state of the art are improved. To solidly argue the article's contribution, choosing a benchmark against which a clear improvement can be demonstrated is essential. 

Other important aspects to improve are the following:

·         If possible, improve the quality of Figures 6, 7, and 8.

·         The conclusions must be rewritten, clearly identifying the contribution. It is also suggested to include some quantitative results.

Reviewer 3 Report

Comments and Suggestions for Authors

This work addressed the dynamic risk assessment of voltage violations in distributed generation, with the verification on the IEEE-33 system. Clearly, the authors exhibit a deep and erudite understanding of the knowledge related. But a consideration of other issues, treated in some detail below, raises legitimate and serious concerns. 

1. The Abstract looks redundant in presenting the significance of this work.

2. The natural environment factors include more factors than those addressed in this work. It will be accurate to specify what the focused environment factors are.

3. As the authors stated, the choice of the kernel function affects the results of KDE. Why the unimodal kernel function centered at 0 is chosen? Any insights or results to show the influence? 

4. Is that sound that only the global horizontal radiation is adopted as the conditional variable for PV, and wind speed for WT? 

5. Why the standard deviation is set at 20% of the expected value in Fig.4?

6. The two verification cases in this work seem not accurate in the input and output. More solid verification is suggested to support the method proposed. 

Comments on the Quality of English Language

The writing looks good. The quality of the English language is acceptable. 

Round 2

Reviewer 2 Report

Comments and Suggestions for Authors

All my concerns were resolved.

Reviewer 3 Report

Comments and Suggestions for Authors

The Authors have responded to my comments properly. No further questions exist. 

Comments on the Quality of English Language

English writing is fine. Meanwhile, I would suggest the Authors scrutinize this paper again, to modify some minor mistakes.